# Unraveling TAFRO Syndrome: An In-Depth Look at the Pathophysiology, Management, and Future Perspectives

**DOI:** 10.3390/biomedicines12051076

**Published:** 2024-05-13

**Authors:** Juan Carlos Caballero, Nazaret Conejero, Laura Solan, Francisco Javier Diaz de la Pinta, Raul Cordoba, Alberto Lopez-Garcia

**Affiliations:** 1Department of Hematology, Fundacion Jimenez Diaz University Hospital, 28040 Madrid, Spain; nazaret.conejero@quironsalud.es (N.C.); laura.solan@quironsalud.es (L.S.); raul.cordoba@fjd.es (R.C.); alberto.lgarcia@quironsalud.es (A.L.-G.); 2Health Research Institute IIS-FJD, 28040 Madrid, Spain; 3Department of Pathology, Fundacion Jiménez Diaz University Hospital, 28040 Madrid, Spain; fjavier.diazp@quironsalud.es

**Keywords:** TAFRO, thrombocytopenia, anasarca, fever, renal failure, reticulin fibrosis, organomegaly

## Abstract

TAFRO syndrome is a rare and aggressive inflammatory entity characterized by thrombocytopenia, anasarca, fever, renal failure, reticulin fibrosis, and organomegaly. This entity supposes a diagnostic and therapeutic challenge due to its significant overlap with Castleman’s disease. However, distinct clinical and histological features warrant its classification as a separate subtype of idiopathic multicentric Castleman’s disease (iMCD). While recent modifications have been made to the diagnostic criteria for iMCD, these criteria lack specificity for this particular condition, further complicating diagnosis. Due to its inflammatory nature, several complex molecular signaling pathways are involved, including the JAK-STAT pathway, NF-kB, and signal amplifiers such as IL-6 and VEGF. Understanding the involvement of immune dysfunction, some infectious agents, genetic mutations, and specific molecular and signaling pathways could improve the knowledge and management of the condition, leading to effective treatment strategies. The current therapeutic approaches include corticosteroids, anti-IL6 drugs, rituximab, and chemotherapy, among others, but response rates vary, highlighting the need for personalized strategies. The prognosis is uncertain due to diagnostic difficulties, emphasizing the importance of early intervention and appropriate targeted treatment. This comprehensive review examines the evolving landscape of TAFRO syndrome, including the pathophysiology, diagnostic criteria, treatment strategies, prognosis, and future perspectives.

## 1. Introduction

TAFRO syndrome, a rare disease first described in 2010 by Takai et al., is characterized by a constellation of symptoms including thrombocytopenia, anasarca (edema, pleural effusion, and ascites), fever, renal failure, reticulin fibrosis, and organomegaly (hepatosplenomegaly and lymphadenopathy) [1]. Although its annual incidence in Japan is estimated to be between 110 and 502 cases per million individuals [2], TAFRO syndrome may have been previously underdiagnosed due to its rarity and lack of awareness. A high clinical suspicion is required for identification, as well as appropriate diagnostic methods.

To fully understand TAFRO syndrome, it is essential to explore its relationship with Castleman’s disease (CD), a group of lymphoproliferative disorders with overlapping clinical features [3]. While CD was first described in 1956, it was not until 2010 that Takai et al. established TAFRO syndrome as a variant of idiopathic multicentric Castleman’s disease (iMCD) [4]. Notably, the geographic distribution of CD variants varies significantly, with Human herpes virus 8 (HHV-8) associated cases being more common in Western countries and idiopathic cases more prevalent in Japan. This highlights the importance of regional expertise and classification systems in the diagnosis and management of this complex disease [5,6]. By elucidating the connections between TAFRO and CD, we can gain a deeper understanding of the underlying pathophysiology and develop more effective treatment strategies for this rare and challenging disorder.

The classification of CD has been established on the basis of the distribution of enlarged lymph nodes into two variants, the hyalinovascular (HV) and the plasmacytic (PC) differentiation variant, although the distinction is often diffuse with overlapping features. While the hyalinovascular variant consists of an interfollicular vascular proliferation, the PC variant consists of a proliferation of PC styloid cells in interfollicular areas [7]. Thus, the disease has been classified into two main groups, the unicentric variety, which characteristically affects one or more lymph nodes with HV differentiation, and the multicentric variant, with PC differentiation, multifocal involvement, and, on many occasions, systemic symptoms [4].

Moreover, two subgroups of multicentric Castleman’s disease (MCD) could be outlined, excluding the variety associated with POEMS (polyneuropathy, organomegaly, endocrinopathy, M-protein, and skin changes) syndrome, which we will not discuss in this review as it is not our purpose. The first subgroup corresponds to the one associated with HHV-8. This subgroup is characteristically found in immunocompromised patients, in most cases patients with HIV (human immunodeficiency virus) infection [7]. The other subgroup involves those of unknown etiology, known as iMCD or iMCD-NOS (not otherwise specified). Although the origin of this disorder is still under debate, a clear relationship with elevated levels of interleukin 6 (IL-6) is recognized, which lays the basis for therapeutic management [4]. TAFRO syndrome is a systemic disease with features resembling iMCD and classically unrelated to HHV-8. Thus, TAFRO syndrome is classified as an invasive clinical subtype of iMCD [1].

### 1.1. Pathophysiology and Biology of TAFRO

TAFRO syndrome, characterized by its aggressive inflammatory nature, is a complex disorder that involves the intricate interplay of several signaling pathways. Understanding these pathways is crucial for unraveling the disease’s pathophysiology and developing effective treatment strategies.

The unknown etiology of these iMCD cases presents a significant challenge to the design of appropriate treatment regimens. Possible etiologies in these patients include a virus other than HHV-8, paracrine cytokine secretion by a small population of neoplastic cells, or autoinflammatory mechanisms. Despite the uncertainty of their etiology, several inflammatory pathways are known to be involved and activated in their development [8].

### 1.2. JAK-STAT Pathway

The JAK-STAT pathway acts as one of the main conductors, orchestrating the inflammatory response through various cytokines, including IL-6, IL-27, IL-10, and TNF-α. IL-6 plays a pivotal role, binding to its receptor and activating JAK1. This activation triggers a cascade of events, ultimately leading to B cell and plasma cell differentiation, as well as the production of acute-phase reactants. Additionally, JAK interacts with the PI3K/AKT/mTOR pathway, further amplifying the inflammatory response. The PI3K/AKT/mTOR pathway, known to be involved in both autoimmune and malignant diseases, plays a diverse role in both normal physiological processes and TAFRO’s pathogenesis [9]. Activated by various factors, it regulates cell proliferation, differentiation, protein synthesis, and cellular metabolism. This pathway’s collective activation in TAFRO lymph nodes, along with VEGF expression and CD8-T-cell activation, contributes to the disease’s progression. This excessive inflammatory response within immune cells can contribute to changes in the structure of lymph nodes and the development of diverse clinical symptoms [10].

Another molecule involved in the activation of the PI3K/AKT/mTOR pathway is the insulin-like growth factor binding protein-1 (IGFBP-1), which plays a role in controlling cell proliferation, adhesion, apoptosis, migration, and invasion by interacting with cell-surface molecules [10]. Sumiyoshi et al. identified molecular differences between TAFRO and iMCD that have potential diagnostic and prognostic value in distinguishing these different types of idiopathic multicentric Castleman’s disease; they found increased activation of the mTOR pathway in TAFRO compared to iMCD-NOS, which may elevate IGFBP-1, identifying IGFBP-1 as a promising new diagnostic and prognostic biomarker, with significantly higher levels in TAFRO than in iMCD-NOS, as well as in patients with other autoimmune diseases [11,12].

### 1.3. NF-kB Pathway

The NF-kB pathway acts as a critical regulator at the cellular level. Normally inactive, it becomes activated by inflammatory signals like IL-1β, IL-17, and TNF-α. This activation, in turn, initiates the transcription of the IL-6 gene and regulates VEGF secretion, further fueling the inflammatory cascade. IL-6, a potent inflammatory cytokine, plays a key role in regulating hematopoiesis, inflammation, and immune response [10]. VEGF, on the other hand, promotes vascular permeability and angiogenesis. In TAFRO, VEGF contributes to increased vascular leakage and endothelial cell damage, while also stimulating IL-6 production in the bone marrow. This creates a vicious cycle, where IL-6 and VEGF reinforce each other’s production, perpetuating the inflammatory response and tissue damage [13].

### 1.4. IL-6 and VEGF

There are some classic symptoms that are associated with elevated serum IL-6 levels, such as thrombocytosis and polyclonal hypergammaglobulinemia. It is noteworthy that these signs do not usually appear in TAFRO syndrome, which could be due to the fact that the elevation of this cytokine is one of the accompanying factors that lead to a poor prognosis but is not one of the primary triggers. Within the bone marrow, specialized cells called stromal cells release VEGF, which in turn triggers the production of IL-6. This creates a self-reinforcing loop in both lymph nodes and bone marrow, where IL-6 and VEGF fuel each other’s production, leading to a sustained inflammatory response and the formation of long-lasting antigenic targets [7,10].

### 1.5. Autoimmune Dysfunction

Nearly half of TAFRO patients exhibit signs of autoimmune dysfunction, as evidenced by the presence of specific antibodies targeting various tissues and components. These antibodies, such as those associated with rheumatoid arthritis, platelets, the thyroid gland, and Sjögren’s syndrome (SS), highlight the potential involvement of the immune system in TAFROs development. Several authors emphasize the importance of determining the presence of anti-SSA/Ro60 antibodies in patients with TAFRO [10,14]. The pathophysiological process by which the disease develops is illustrated in Figure 1.

## 2. Diagnosis

### 2.1. Masaki’s Criteria

Prior to the introduction of Masaki’s diagnostic criteria in 2015 [15], TAFRO syndrome was a poorly understood and often misdiagnosed condition. The lack of standardized diagnostic criteria led to significant delays in diagnosis and treatment, which contributed to high rates of morbidity and mortality [8]. Masaki’s criteria provided a much-needed framework for the diagnosis of TAFRO syndrome.

Major criteria include anasarca (pleural effusion, ascites, and edema); thrombocytopenia (<100,000/μL, no myelosuppression); and systemic inflammation (fever > 37.5 °C, CRP ≥ 2 mg/dL). Minor criteria include CD-like features on lymph node biopsy; reticulin myelofibrosis/increased megakaryocytes in bone marrow; mild organomegaly (hepatomegaly, splenomegaly, and lymphadenopathy); and progressive renal insufficiency. All three major categories (anasarca, thrombocytopenia, and systemic inflammation) and at least two of four minor categories are required for diagnosis. Some pathologies must be ruled out in order to establish a diagnosis. These include other malignancies such as lymphomas or myelomas, autoimmune diseases such as Systemic Lupus Erythematosus (SLE) or SS, infectious diseases, POEMS syndrome, hepatic cirrhosis, or thrombotic thrombocytopenic purpura/hemolytic-uremic syndrome [16]. The diagnostic protocol that we have developed can be seen in Figure 2.

The criteria are not without limitations and are not specific to TAFRO syndrome. Some patients with other conditions, such as MCD without TAFRO syndrome, may meet the criteria. Also, some patients with TAFRO syndrome may not meet all of the criteria. Despite these limitations, Masaki’s criteria have made a significant contribution to the diagnosis and management of TAFRO syndrome [8].

### 2.2. Histology

First, lymph node biopsy is useful because TAFRO syndrome is a diagnosis of exclusion. It helps to rule out other conditions that may present similar clinical symptoms such as Epstein–Barr virus (EBV)-related hyperplasia, autoimmune lymphadenopathy, HHV-8-MCD, POEMS-MCD, lymphomas, or follicular dendritic cell sarcoma.

Once iMCD diagnosis is established, it is mandatory to differentiate between iMCD-NOS and iMCD-TAFRO, although this relies more on clinical criteria than histological ones because it may be not possible to differentiate between both entities based on histology. It is worth noting that two groups of criteria have been proposed: Iwaki 2016, where a histological lymph node sample would be necessary; and Masaki 2019, where this is considered a minor criterion since it cannot be performed sometimes, or patients present small-volume lymphadenopathy. The main histological findings include regressed germinal centers, follicular dendritic cell prominence, hypervascularity of germinal centers and interfollicular area, and an increase in plasmacytoid cells in the interfollicular area and germinal center hyperplasia. According to a recently published multidisciplinary consensus document, these five key features must be graded in a four-tiered scheme (0–3) [17].

Furthermore, three histopathologic subtypes of iMCD have been proposed: hypervascular (formerly hyaline-vascular), mixed, and PC pathology [18]. Generally, iMCD patients with TAFRO syndrome present hypervascular or mixed subtypes. However, some patients do not present these findings, or different subtypes may be encountered on subsequent biopsies or simultaneous biopsies of separate samples, so this should be considered as a spectrum of findings, and the reliability and clinical utility of subtyping is currently unclear. When iMCD-TAFRO is suspected, bone marrow biopsy and/or aspirate is recommended, and the main histological findings are reticulin fibrosis and increased megakaryocytes [19]. Some of these histopathological features are shown in Figure 3.

Regarding kidney histology, most cases of iMCD-TAFRO present membranoproliferative glomerulonephritis-like changes with double contour of glomerular basement membrane and narrowing of glomerular tuft due to endothelial cell swelling. Thrombotic microangiopathy (TMA) is a condition that presents with thrombocytopenia, anemia, and renal failure, which are findings that may suggest clinical overlap with TAFRO syndrome. Furthermore, some cases with histological changes to those observed in thrombotic microangiopathy have been reported [20]. However, recent reviews have established that typical histological criteria of TMA, such as fibrin thrombi, fibrinoid necrosis of glomerular capillaries, mucoid intimal thickening, and erythrocyte fragmentation, are not generally found in TAFRO syndrome [21].

### 2.3. Other Diagnostic Features

IL-6 may function as an indicator of proliferation since it is thought to be the cause as it is a regulator of mesangial proliferation. In patients with clinical suspicion and renal involvement, a renal biopsy may be useful. Chronic peribronchial and periarterial inflammation accompanied by the formation of lymphoid follicles can be observed in the lungs. In addition, fibrotic changes and eosinophilic infiltration may also be seen. At the cutaneous level, glomeruloid hemangiomas can be observed. Thymic, epithelial, and lymphocytic hyperplasia with septal fibrosis may be observed in the thymus [22].

As for the pathophysiology of thrombocytopenia, it is not very clear, but Audia et al. have observed how splenic macrophages phagocytize platelets, with the consequent development of antiplatelet antibodies that later generate the picture of thrombocytopenia [23].

The development of anti-platelet IgG has been reported and all this is presumed to be the cause of hypersplenism and thrombocytopenia. In relation to pulmonary involvement, although less described, lymphoplasmacytic proliferation can be observed, especially in the alveolar area adjacent to the perilymphatic stroma [24]. A relevant fact that has been analyzed by some authors is the relationship between TAFRO syndrome and SS. It has been observed that in some patients, there is a concurrence of the two pathologies, with some authors going so far as to state that TAFRO could even be a severe manifestation of SS.

Although it is very useful for diagnosis, in recent years, a confirmatory biopsy showing data compatible with CD is no longer considered an indispensable criterion, since there are patients with comorbidities or severe thrombocytopenia that contraindicate its performance. For this reason, in Masaki’s criteria, histological confirmation is not a fundamental criterion but is considered secondary [14].

A key point when diagnosing this entity is to rule out other pathologies that can simulate a TAFRO, such as SLE or POEMS. It may be a matter of debate to add SS to this list of diseases. Nevertheless, the presence of anasarca, extratubular involvement of the kidney, or cytopenias secondary to myelofibrosis, leads to a more focused consideration of the case towards TAFRO, as these symptoms are rare in SS [14]. Due to the overlapping of symptoms at different levels and systems, the need for several specialists for the diagnosis of the disease is usual [10].

Research by Morit et al. identified various antibodies targeting different platelet components in TAFRO patients, including glycoproteins, human leukocyte antigens, and human platelet antigens. This finding strongly suggests that the low platelet count (thrombocytopenia) observed in TAFRO syndrome might be caused by the production of these autoantibodies, which attack and destroy platelets [25].

### 2.4. Role of Infections

While HHV-8 is the primary culprit behind HHV-8-positive MCD, TAFRO syndrome, despite sharing similar clinical features and pathology, has not been linked to any human herpesvirus infection. Initial studies suggest that patients with iMCD often harbor various pathogens, including EBV, Human herpes virus 6, hepatitis B virus, cytomegalovirus (CMV), *Toxoplasma gondii*, or *Mycobacterium tuberculosis*. Interestingly, case reports have documented TAFRO syndrome co-occurring with EBV and CMV infections [10].

In this context, it is noteworthy to highlight the distinctions between TAFRO and Kaposi’s sarcoma-associated herpesvirus-inflammatory Cytokine Syndrome (KICS); a syndrome that may occasionally share overlapping features and is indeed associated with HHV-8. KICS is a serious illness that can cause fever, fatigue, and respiratory failure, among others. Both TAFRO and KICS lack a definitive diagnostic test and rely on clinical presentation and exclusion of other causes. KICS typically presents with more severe symptoms and a higher mortality rate compared to TAFRO. Additionally, newly described histopathological features of KICS (sinusoidal dilatation and atypical plasmablasts) are not usually seen in TAFRO [26,27,28]. Table 1 summarizes the key distinctions between iMCD-NOS, TAFRO, and KICS.

The potential involvement of EBV is being investigated in the development of angiogenesis in CD. The rationale for this investigation stems from EBV’s established role in various cancers, including lymphomas, where it can influence cellular growth factors like IL-6 and potentially angiogenesis through interleukin-8 [29]. The observed association between EBV presence and CD, particularly in germinal centers of lymphoid tissue, suggests a potential link. In these areas, EBV could theoretically stimulate IL-6 production, thereby contributing to CD development. Furthermore, EBV might influence angiogenesis in the early stages of CD, similar to its suspected role in nasopharyngeal carcinoma, by inducing IL-8 production via its latent membrane protein-1. These findings raise the possibility that targeting angiogenesis could be a promising therapeutic approach for CD, particularly in lesions with high angiogenic activity [30].

To investigate the potential role of bacterial infections in TAFRO syndrome, Kageyama et al. analyzed DNA sequences extracted from liver samples of TAFRO patients. Interestingly, they identified sequences closely resembling those of *Campylobacter jejuni*. However, attempts to directly detect the bacteria using immunohistochemistry and electron microscopy were unsuccessful. These findings suggest that *Campylobacter jejuni* might indirectly contribute to TAFRO by triggering the production of autoantibodies through a process called cross-immune reaction. This, in turn, could lead to an uncontrolled release of inflammatory molecules, known as a cytokine storm [31].

**Table 1 biomedicines-12-01076-t001:** Comparative table between iMCD-NOS, TAFRO, and KICS. Inspired by the table prepared by Dispenzieri et al. [32]. PN, peripheral neuropathy; AIHA, autoimmune hemolytic anemia; PNP, paraneoplastic pemphigus; ITP, immune thrombocytopenic purpura; LANA, latency-associated nuclear antigen; HAART, highly active antiretroviral therapy; LPS, lymphoproliferative syndromes; PEL, primary effusion lymphoma; PBL, plasmablastic lymphoma.

	iMCD-NOS	iMCD-TAFRO	KSICS
**Age**	Fifth to sixth decade	Fifth decade	Fourth to fifth decade
**Clinical presentation**	B symptoms and occasional PN	B symptoms and anasarca	Fever, anasarca, multiorgan failure
**Lymphadenopathy**	Very frequent	Very frequent	May be present (reactive)
**Organomegaly**	May be present	Very frequent Supports diagnosis	May be present
**Body effusion**	Infrequent	Must be present	Very frequent
**Abnormal inflammatory markers**	Release of pro-inflammatory cytokines	Release of pro-inflammatory cytokines	Release of pro-inflammatory cytokines Evidence of HHV-8 viral activity
**Cytopenia**	May be present Sometimes thrombocytosis	Thrombocytopenia must be present	May be present
**Renal dysfunction**	Frequent	Very frequent	May be present
**Autoimmune phenomena**	Very frequent: AIHA, PNP, ITP, interstitial lung disease	Infrequent	Infrequent
**Pathologic features (lymph node)**	Usually, PC variant	Usually mixed or hypervascular type	Exclusion of MCD. Reactive plasmacytosis and node hyperplasia. KSHV-infected plasma cells. KSHV-LANA may be present
**Therapy**	IL-6-targeted therapy; rituximab; systemic therapies	Same as iMCD, but also calcineurin inhibitors	Rituximab, doxorrubicin, HAART, and support. Valganciclovir and Zidovudine may be useful
**Clinical course**	Variable	Very aggressive	Very aggressive 60% mortality
**Risk for lymphoma**	High	Mild	Very high risk of KSHV-related LPS, (PEL, PBL…)

### 2.5. Mutations

Recent studies suggest that acquired gene mutations may contribute to the development of TAFRO syndrome. Researchers have identified several potential mutations, including amplifications in the *ETS1*, *PTPN6*, and *TGFBR2* genes [10]. Yoshimi et al. identified specific mutations, *MEK2^P128L^* and *RUNX1^G60C^*, and the last one is known to enhance cell self-renewal in patients with TAFRO syndrome. These mutations may contribute to the disease by activating a cell signaling pathway (MAPK) and potentially offer new therapeutic targets. Additionally, a somatic *DNMT3A^L295Q^* mutation commonly associated with various tumors has been identified in TAFRO patients. As research into the genetic basis of TAFRO continues, these findings hold promise for the development of targeted therapies in the future [33].

## 3. Treatment Strategy

### 3.1. Corticosteroids

It is common for most patients to have a poor outcome and to be refractory to treatment. Some patients respond to immunosuppressive drugs, glucocorticoids, cyclosporine A (CsA), tocilizumab (TCZ), or rituximab [34]. Corticosteroids, as immunosuppressive agents, effectively address both acute and chronic inflammation by reducing the transcription of proinflammatory cytokines, chemokines, adhesion molecules, and key enzymes involved in the inflammatory process (e.g., IL-2, IL-6, and TNF-α). They serve as the primary treatment for conditions such as iMCD, including TAFRO syndrome. Despite being the most commonly used first-line drug, corticosteroids have limited action, with few patients having a response [34]. Relapses are common, so prompting combination therapy with other medications must be suggested. In cases where corticosteroids prove ineffective, alternative drugs like anti-IL-6 and chemotherapeutic agents such as anakinra, bortezomib, and thalidomide are often administered. However, limited data exist on the prognosis and response to these treatments [35]. Studies evaluating these regimens often involve case reports and retrospective analyses, which are inherently susceptible to bias in patient selection. This makes it challenging to draw definitive conclusions about their effectiveness [10]. The most recommended dose of corticosteroids by different authors is high doses of milligrams per kilogram of body weight per day (mg/kg/day). In patients with metabolic syndrome, impaired glycemic control, or adverse effects secondary to steroids, other immunosuppressants used as steroid-sparing agents may be the best option [34].

### 3.2. Anti-IL-6 Therapies

Despite these limitations, some medications have emerged as potential options for TAFRO treatment. TCZ, with an effectiveness of around 50%, and siltuximab are frequently considered for first-line therapy, while rituximab, CsA, sirolimus, tacrolimus, thalidomide, lenalidomide, bortezomib, and lymphoma chemotherapy regimens may be used as second-line options [34].

TCZ is a humanized monoclonal antibody directed towards the IL-6 receptor. By binding to both soluble and membrane-bound interleukin-6 receptors, TCZ impedes the proinflammatory actions of IL-6. Studies by Nishimoto et al. have confirmed its efficacy in initiating and sustaining remission in patients with iMCD [36].

Siltuximab, a human–murine chimeric monoclonal antibody with a high affinity for IL-6 [37], demonstrated efficacy in a Phase III clinical trial when combined with optimal supportive care, surpassing the outcomes of optimal supportive care alone. It is currently approved for MCD treatment only in North America and Europe. In contrast, TCZ, a humanized IL-6 antagonist that blocks transmembrane signaling of IL-6, is approved for MCD treatment in Japan and globally for rheumatoid arthritis [35]. The most commonly used doses of these two drugs are those recommended in their technical data sheets, which can be seen in more detail in Figure 4.

Renal toxicity may be a limiting factor for its use. It is possible that a combination of drugs with steroids and other immunosuppressants may be effective and useful in cases of severe TAFRO. Beyond corticosteroids and anti-IL-6 therapies, the responses are significantly lower. Published case reports show a better disease response when early immunosuppression is established [34].

### 3.3. Thalidomide

Thalidomide, an immune modulator, inhibits various cytokines, including TNF-α, IL-1, IL-6, IL-12, and VEGF, while also promoting T-cell activity. It has proven effective in inducing remission, reducing IL-6 levels, and lowering CRP in iMCD patients [35].

### 3.4. Rituximab

Rituximab, a chimeric monoclonal antibody targeting CD20, approved for non-Hodgkin lymphoma, is commonly used as first- or second-line therapy for MCD. While it may be considered for iMCD cases linked to immune-related disorders, rituximab’s efficacy might be partial [35]. Some studies show a more durable response than that achieved with TCZ in terms of time until the next treatment. Despite this, overall survival does not seem to be improved when confronted with the different therapeutic options [2].

### 3.5. Chemotherapy

Different authors have used multiple different chemotherapy regimens with modest results. There is no consensus on the best chemotherapeutic treatment for patients with TAFRO. CHOP (cyclophosphamide, hydroxydaunorubicin, oncovin, vincristine, and prednisone) has poor results when used in this context, suggesting the idea that more aggressive drug combinations may be necessary for the treatment of these patients [38,39].

Iwaki et al. described a case of a patient with TAFRO syndrome who was refractory to corticosteroids, rituximab, and siltuximab. The patient achieved complete remission after three cycles of bortezomib (velcade)-dexamethasone-thalidomide-adriamycin-cyclophosphamide-etoposide-rituximab (VDT-ACE-R) chemotherapy. However, the patient relapsed 15 and 16 months after the first and second cycles, respectively, while on siltuximab maintenance therapy. The patient received a third cycle of VDT-ACE-R and achieved another complete remission, receiving maintenance therapy with intravenous immunoglobulin (IVIG) and sirolimus reaching remission for 21 months. Chemotherapy for inducing remission in TAFRO syndrome that is refractory to other therapies may be helpful in some cases. Additionally, the combination of IVIG and sirolimus may be an effective maintenance therapy for preventing relapse in patients with TAFRO syndrome [8]. The severity scale based on the criteria proposed by Masaki et al., with therapeutic implications, is shown in Table 2.

### 3.6. Additional Treatments

Shirai et al. reported two cases in 2018 following treatment with Tacrolimus. Both patients responded with a disease remission lasting over two years [40]. Additionally, treatments like plasma exchange, hematopoietic stem cell transplantation, and the thrombopoietin receptor agonist anakinra, might offer potential benefits for specific patients, but they typically fall under third-line therapy considerations. It is important to remember that the choice of treatment depends heavily on individual patient characteristics and their response to previous therapies [10]. Further research is needed to evaluate the efficacy of these treatment options in a larger cohort of patients.

As reported in some cases in the literature, in patients with prolonged immunosuppression, special caution should be taken with opportunistic diseases, since they can have a poor evolution in patients with these conditions. Some patients may benefit from the use of prophylactic antimicrobials, although there are still no general recommendations in this regard [41]. The proposed therapeutic algorithm is shown in Figure 4. It includes treatment recommendations based on the severity scale of the disease. In the case of intermediate or “slightly severe” cases, the treatment decision should be individualized, based on the patient’s characteristics and the judgment of the physician.

**Figure 4 biomedicines-12-01076-f004:**
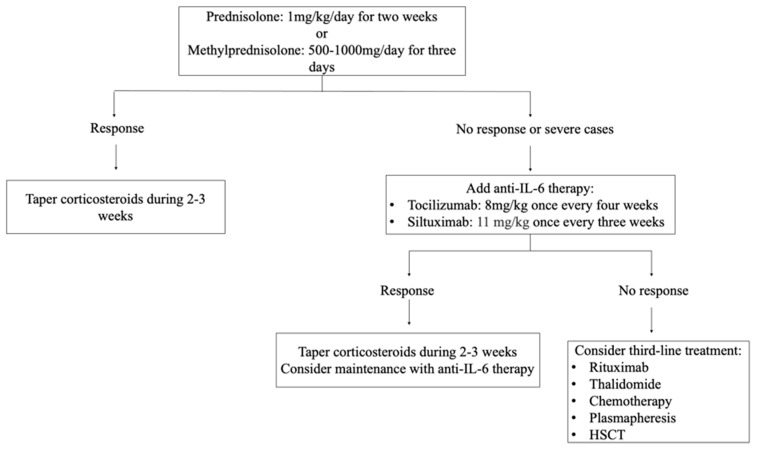
Proposed algorithm for the treatment of TAFRO syndrome. HSCT: hematopoietic stem cell transplantation.

## 4. Prognosis

Some authors have defined some prognostic factors predictive of poor overall survival. These include age > 60 years and D-dimer above 18 μg/dL. After 24 months of disease progression, global survival rates drop rapidly, and up to one-third of patients die [2].

An important cause of high mortality in TAFRO syndrome is the difficulty in diagnosis, being notoriously challenging to obtain confirmatory biopsies in some cases and with a rapid progression of the disease. The 5-year survival is approximately 66.5%. A key factor in the low survival rates is the delay in establishing treatment, as diagnosis is sometimes missed [10].

According to one of the largest case series published to date in 2016 by Iwaki et al., the performance status at diagnosis is generally poor, with an ECOG greater than one in almost 80% of patients and a follow-up from diagnosis to the last date of only 9 months [8].

## 5. Future Perspectives

Unfortunately, the rarity of this illness has constrained the investigation of novel therapeutic approaches in recent times. A significant, and possibly the most up-to-date, development has been to recognize a TAFRO syndrome subtype with a notably poor prognosis, distinguished from being merely a subgroup or an overlapping category. This study discerns TAFRO and iMCD as having divergent clinical manifestations. TAFRO could be an autoimmune disease characterized by the presence of anti-SSA antibodies with an intense phenotype, which implies the necessity to devise future therapeutic interventions targeting this antibody due to its typical resistance to tocilizumab therapy [42].

Over the last ten years, there has been considerable advancement in our comprehension of iMCD. The challenge of distinguishing between iMCD-TAFRO and iMCD-NOS based on histological and clinical features is recognized in the literature. Therefore, there is a growing need to improve diagnostic strategies by exploring additional biomarkers or diagnostic imaging modalities. Emphasis on the development and validation of such tools could significantly improve the accuracy and efficiency of diagnosis and differentiation of these subtypes of idiopathic multicentric Castleman’s disease [10].

The development of diagnostic criteria and therapeutic algorithms has been instrumental, alongside the Food and Drug Administration’s approval of siltuximab. Notably, the approval of siltuximab occurred before the formalization of diagnostic and therapeutic standards for iMCD. This chronology may account for the apparent discrepancy in treatment efficacy observed in clinical studies when retrospectively applied to the now-established guideline. Further studies are warranted to unravel the origins and development of iMCD, specifically investigating the interplay between environmental exposures, genetic predispositions, and other individual-specific factors that may precipitate its onset. Moreover, the contribution of immune system abnormalities to the disease, as well as the underlying reasons why some patients do not respond to treatment or experience a return of the disease, remains to be fully elucidated. Multidisciplinary teams are needed.

## 6. Conclusions

TAFRO syndrome presents diagnostic difficulties due to its complex etiology and diverse clinical manifestations. A multidisciplinary approach is crucial for accurate diagnosis and optimal management. Elucidating the underlying inflammatory pathways holds potential for targeted therapies, yet the specific mechanisms require further investigation. Standardized diagnostic criteria improve early recognition, but limitations remain in differentiating TAFRO from similar presentations. Corticosteroids are the mainstay of treatment, but variable responses necessitate alternative options including anti-IL6 agents, rituximab, or chemotherapy. A treatment algorithm based on corticosteroid response and disease severity is proposed. The prognosis hinges on timely diagnosis and targeted therapy. Future research should focus on unraveling the pathogenesis, refining diagnostics, and developing targeted therapies to improve patient outcomes. Despite ongoing challenges, collaborative efforts offer promise for improved understanding and patient care.

## Figures and Tables

**Figure 1 biomedicines-12-01076-f001:**
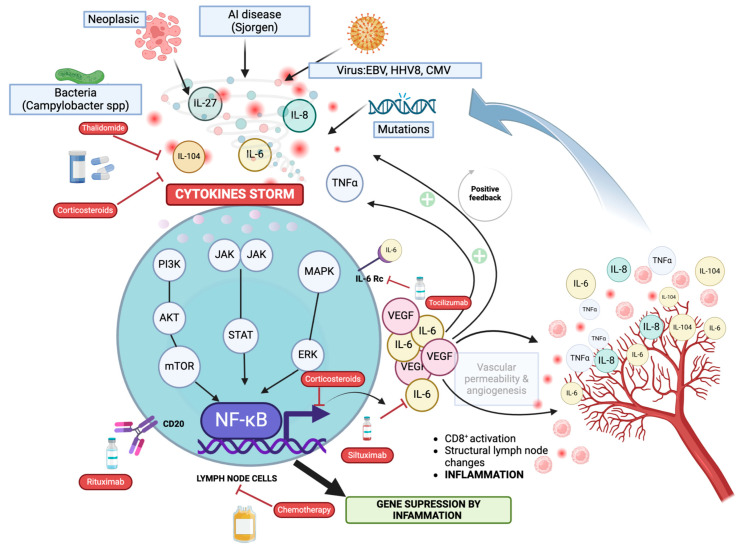
Pathophysiology, etiological factors, and potential therapeutic targets in TAFRO syndrome. EBV: Epstein—Baar virus; HHV8: Human herpes virus 8; CMV: cytomegalovirus; AI: autoimmune; TNF: tumor necrosis factor; IL: interleukin; MAPK: MAP kinase; VEGF: vascular endothelial growth factor; PI3K: phosphoinositide 3-kinase; AKT: protein kinase B; NFkB: nuclear factor kappa-light-chain-enhancer of activated B cells; Rc: receptor.

**Figure 2 biomedicines-12-01076-f002:**
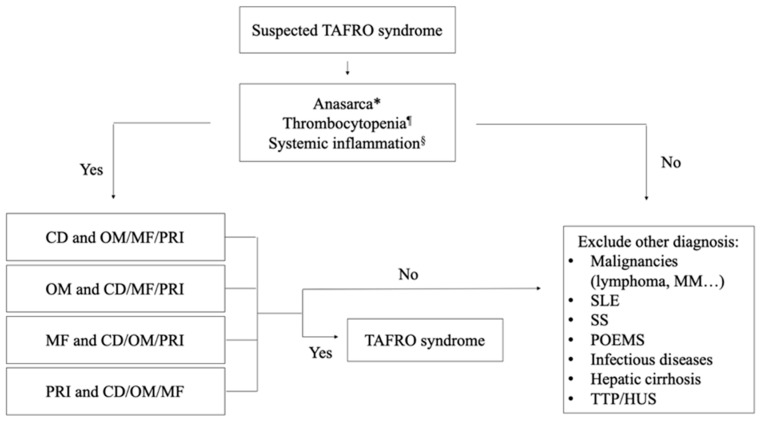
Proposed algorithm for the diagnosis of TAFRO syndrome. CD: Castleman-like features on lymph node biopsy; OM: organomegaly; MF: myelofibrosis; PRI: progressive renal insufficiency; SLE: systemic lupus erythematosus; SS: Sjögren’s syndrome; POEMS: polyneuropathy, organomegaly, endocrinopathy, monoclonal protein, and skin changes; TTP/HUS: thrombotic thrombocytopenic purpura/hemolytic uremic syndrome. * Pleural effusion, ascites and/or edema. ^¶^ <100,000/μL and no myelosuppression. ^§^ Fever > 37.5 °C and CRP ≥ 2 mg/dL.

**Figure 3 biomedicines-12-01076-f003:**
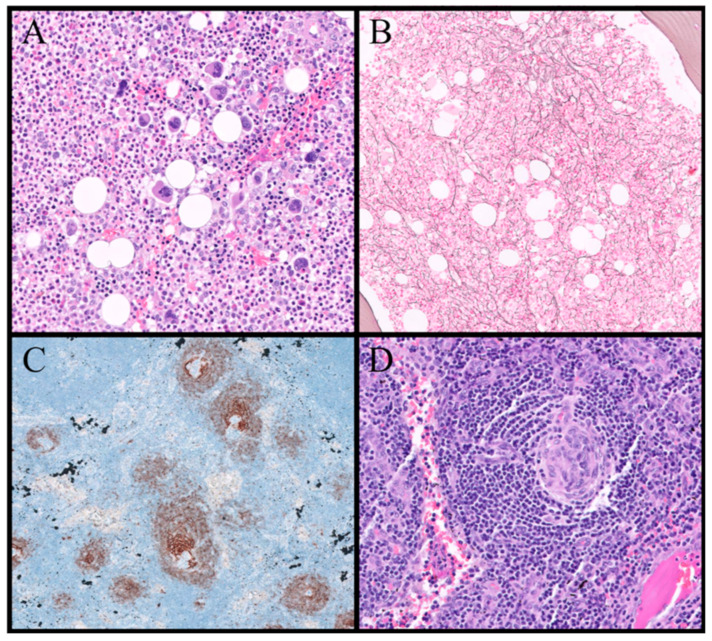
Histologic specimens of bone marrow and lymph node in a patient with TAFRO syndrome. (**A**) Bone marrow, Hematoxylin-eosin (HE). Megakaryocytic hyperplasia with signs of dysplasia (×5). (**B**) Bone marrow. Reticulin fibrosis. Reticulin staining (×1). (**C**) Lymph node. Hyperplasia of dendritic cells labeled with CD23 (×5). (**D**) Lymph node, HE. Vessels entering the germinal center and lymphocytes in a single file. Typical “lollipop” lesions (×5).

**Table 2 biomedicines-12-01076-t002:** Severity scale based on the criteria proposed by Masaki et al. PE: physical examination; CPR: C-reactive protein; GFR: glomerular filtration rate.

Severity Score (Points)	Anasarca	Thrombocytopenia	Fever and/or Inflammation	Renal Insufficiency
1	Pleural effusion, ascites, or pitting edema on PE	Platelet counts < 100,000/μL	Fever ≥ 37.5 °C but <38.0 °C or CRP ≥ 2 mg/dL but <10 mg/dL	GFR < 60 mL/min/1.73 m^2^
2	Two of the above	Platelet counts < 50,000/μL	Fever ≥ 38.0 °C but <39.0 °C or CRP ≥ 10 mg/dL but <20 mg/dL	GFR < 30 mL/min/1.73 m^2^
3	Three of the above	Platelet counts < 10,000/μL	Fever ≥ 39.0 °C or CRP ≥ 20 mg/dL	GFR < 15 mL/min/1.73 m^2^ or hemodialysis
**Relationship between Score and Disease Severity**				
0–4 points		Mild		Grade 1
5–6 points		Moderate		Grade 2
7–8 points		Slightly severe		Grade 3
9–10 points		Severe		Grade 4
11–12 points		Very severe		Grade 5

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
