# Peer review of "Unraveling TAFRO Syndrome: An In-Depth Look at the Pathophysiology, Management, and Future Perspectives"

_biomedicines, 2024, doi:10.3390/biomedicines12051076_

Round 1

Reviewer 1 Report

Comments and Suggestions for Authors

The manuscript presents a valuable review of TAFRO syndrome with important insights into diagnosis, pathophysiology, and treatment. To enhance the manuscript further, it would be beneficial to address the following concerns.

1. There is mention of the difficulty in distinguishing iMCD-TAFRO from iMCD-NOS based on histology and clinical presentation alone. More emphasis on developing additional biomarkers or diagnostic imaging could be a valuable addition.

2. It would be insightful to include more molecular studies that correlate enhanced mTOR pathway activation with IGFBP-1 levels. Such studies could validate IGFBP-1 as a potential biomarker for distinguishing iMCD-TAFRO from iMCD-NOS.

Author Response

The manuscript presents a valuable review of TAFRO syndrome with important insights into diagnosis, pathophysiology, and treatment. To enhance the manuscript further, it would be beneficial to address the following concerns.

  1. There is mention of the difficulty in distinguishing iMCD-TAFRO from iMCD-NOS based on histology and clinical presentation alone. More emphasis on developing additional biomarkers or diagnostic imaging could be a valuable addition.

We sincerely appreciate taking the time and dedication to review our manuscript. Your work has greatly improved the quality of our paper.

In response to your suggestions, we included the reference to the difficulty of distinguishing between these two entities. Currently, there is no technique, biomarker, or test that can definitively differentiate between them. We have placed greater emphasis on this fact in the "Future Perspectives" section.

  1. It would be insightful to include more molecular studies that correlate enhanced mTOR pathway activation with IGFBP-1 levels. Such studies could validate IGFBP-1 as a potential biomarker for distinguishing iMCD-TAFRO from iMCD-NOS.

Thank you for suggesting such valuable references. Indeed, this molecule may be useful and is being investigated as a biomarker to differentiate between the two entities. We have added this information and its references in the text under "Pathophysiology and Biology of TAFRO".

Reviewer 2 Report

Comments and Suggestions for Authors

This is a well described report on an interesting, novel and emerging topic. I would like to congratulate the authors on a job well done. My only criticism is that Kaposi Sarcoma Inflammatory Cytokine Syndrome has not been mentioned at all. In fact, TAFRO, MCD and KSICS are clinically indistinguishable, and it would be important to highlight this in the paper. 

I would suggest to the authors to create a table where they can contrast/compare features of TAFRO vs MCD vs KSCIS. This would be very educational and would improve the paper. In addition to comparing and contrasting clinical features, outcome between these 3 entities can be compared as well. 

Suggested literature: 

Clinical Features and Outcomes of Patients With Symptomatic Kaposi Sarcoma Herpesvirus (KSHV)-associated Inflammation: Prospective Characterization of KSHV Inflammatory Cytokine Syndrome (KICS) - PubMed (nih.gov)

A Fatal Case of Kaposi Sarcoma Immune Reconstitution Syndrome (KS-IRIS) Complicated by Kaposi Sarcoma Inflammatory Cytokine Syndrome (KICS) or Multicentric Castleman Disease (MCD): A Case Report and Review - PubMed (nih.gov)

Clinical and pathological features of Kaposi sarcoma herpesvirus-associated inflammatory cytokine syndrome - PubMed (nih.gov)

Comments on the Quality of English Language

minor edits

Author Response

This is a well described report on an interesting, novel and emerging topic. I would like to congratulate the authors on a job well done. My only criticism is that Kaposi Sarcoma Inflammatory Cytokine Syndrome has not been mentioned at all. In fact, TAFRO, MCD and KSICS are clinically indistinguishable, and it would be important to highlight this in the paper. 

I would suggest to the authors to create a table where they can contrast/compare features of TAFRO vs MCD vs KSCIS. This would be very educational and would improve the paper. In addition to comparing and contrasting clinical features, outcome between these 3 entities can be compared as well. 

We are deeply grateful that you took the time to review our manuscript with such dedication. Indeed, as you suggest, the two entities are very difficult to differentiate and share common and overlapping characteristics. We appreciate the suggestion to create a table as you mentioned.

Therefore, we have prepared a comparative table of these three entities with their clinical, analytical, histological, therapeutic, and prognostic characteristics. We have added all the references you suggested, from which we have extracted a lot of information included in the table. We hope you find it to your liking and look forward to further suggestions for improving the quality of our manuscript.

Round 2

Reviewer 2 Report

Comments and Suggestions for Authors

I would like to thank authors for the detailed revision. The paper in its current form is excellently written and is a useful addition to medical literature on TAFRO syndrome. Congratulations!

Comments on the Quality of English Language

minor edits